# Unveiling the gut bacteriome diversity and distribution in the national fish hilsa (*Tenualosa ilisha*) of Bangladesh

A. Q. M. Robiul Kawser[1☯], M. Nazmul Hoque[2☯], M. Shaminur Rahman[3], Tahsin Islam Sakif[4], Tracey J. Coffey[5], Tofazzal Islam[6]*

1 Department of Aquaculture, Bangabandhu Sheikh Mujibur Rahman Agricultural University, Gazipur, Bangladesh, 2 Molecular Biology and Bioinformatics Laboratory, Department of Gynecology, Obstetrics and Reproductive Health, Bangabandhu Sheikh Mujibur Rahman Agricultural University, Gazipur, Bangladesh, 3 Department of Microbiology, Jashore University of Science and Technology, Jashore, Bangladesh, 4 Lane Department of Computer Science and Electrical Engineering, West Virginia University, Morgantown, West Virginia, United States of America, 5 School of Veterinary Medicine and Science, University of Nottingham, Sutton Bonington, United Kingdom, 6 Institute of Biotechnology and Genetic Engineering, Bangabandhu Sheikh Mujibur Rahman Agricultural University, Gazipur, Bangladesh

☯ These authors contributed equally to this work.
* tofazzalislam@bsmrau.edu.bd

**Data Availability Statement:** The 16S rRNA gene amplicon sequencing data are available at the National Center for Biotechnology Information (NCBI) Sequence Read Archive (SRA) under

## Abstract

The field of fish microbiome research has rapidly been advancing, primarily focusing on farmed or laboratory fish species rather than natural or marine fish populations. This study sought to reveal the distinctive gut bacteriome composition and diversity within the anadromous fish species *Tenualosa ilisha* (hilsa), which holds the status of being the national fish of Bangladesh. We conducted an analysis on 15 gut samples obtained from 15 individual hilsa fishes collected from three primary habitats (e.g., freshwater = 5, brackish water = 5 and marine water = 5) in Bangladesh. The analysis utilized metagenomics based on 16S rRNA gene sequencing targeting the V3-V4 regions. Our comprehensive identification revealed a total of 258 operational taxonomic units (OTUs). The observed OTUs were represented by six phyla, nine classes, 19 orders, 26 families and 40 genera of bacteria. Our analysis unveiled considerable taxonomic differences among the habitats (freshwater, brackish water, and marine water) of hilsa fishes, as denoted by a higher level of shared microbiota ($p = 0.007$, Kruskal-Wallis test). Among the identified genera in the gut of hilsa fishes, including *Vagococcus*, *Morganella*, *Enterobacter*, *Plesiomonas*, *Shigella*, *Clostridium*, *Klebsiella*, *Serratia*, *Aeromonas*, *Macrococcus*, *Staphylococcus*, *Proteus*, and *Hafnia*, several are recognized as fish probiotics. Importantly, some bacterial genera such as *Sinobaca*, *Synechococcus*, *Gemmata*, *Serinicoccus*, *Saccharopolyspora*, and *Paulinella* identified in the gut of hilsa identified in this study have not been reported in any aquatic or marine fish species. Significantly, we observed that 67.50% (27/40) of bacterial genera were found to be common among hilsa fishes across all three habitats. Our findings offer compelling evidence for the presence of both exclusive and communal bacteriomes within the gut of hilsa fishes, exhibiting potential probiotic properties. These observations could be crucial for guiding future microbiome investigations in this economically significant fish species.

BioProject accession number PRJNA964437. The accession numbers for all 15 SRA experiments are listed in Table S1.

**Funding:** This research was partially supported with funds from 'BSMRAU Physical Facility and Research Capacity Strengthening Project' under the Ministry of Education, People's Republic of Bangladesh for funding this research, Grant No.: BSMRAU/01/2021. The funders had no role in study design, data collection and analysis, decision to publish, or preparation of the manuscript. There was no additional external funding received for this study.

**Competing interests:** The authors have declared that no competing interests exist.

## 1. Introduction

The hilsa, scientifically named as *Tenualosa ilisha*, holds the esteemed status of being both the national fish and a Geographical Indication (GI) product in Bangladesh. Recognised as an iconic flagship species of the country, it is commonly identified by various names such as ilish, hilsa herring, or hilsa shad. As a member of the Clupeidae family, it shares a close relationship with herring species [1, 2]. Due to its distinct flavor and exceptional taste, the hilsa fish holds significant economic value and experiences high demand worldwide. On a global scale, it is recognized as the most vital commercial transboundary species in the Bay of Bengal. Notably, Bangladesh holds the largest share of this market at 86%, trailed by India at 8%, Myanmar at 4%, with other nations contributing the remaining portion. Moreover, apart from its economic significance, hilsa bears immense sociocultural and religious importance. Its non-consumptive value is estimated to be approximately US$0.36 billion annually [3]. The hilsa-related fishing industry directly supports employment for approximately 0.5 million fishermen. Beyond this, an estimated 2.5 million individuals are involved in various capacities across its extensive value chain [4]. The highly prized and expensive hilsa fish comprising roughly 12% of total fish production, which contributes approximately 1.15% to Bangladesh's Gross Domestic Product (GDP) [5]. It is widely distributed in Southeast and South Asia [6, 7], ranging from China Sea, Bay of Bengal, Arabian Sea, Red Sea to Persian Gulf, and is also found in coastal areas, estuaries, brackish and freshwater rivers [7, 8]. Hilsa, classified as a marine fish, migrates to freshwater environments for spawning. Despite being categorized as an anadromous species, hilsa can be found in most of the main rivers throughout the year. Due to its migratory nature, a uniform or homogeneous stock of hilsa is not typically anticipated [4]. The microbial communities residing in the guts of hilsa fish have the potential to augment the host's metabolic capabilities by positively influencing processes such as nutrient digestion and assimilation. Additionally, these communities may serve a protective role by defending the host against invasive pathogens [9].

The composition of microbial communities in aquatic ecosystems, including fish skin microbiota, is influenced by various factors such as water quality, environmental conditions, and host-specific factors. Previous studies reported a high abundance of *Proteobacteria*, *Bacteroidetes* and *Actinobacteria* in the skin of anadromous fish, including Atlantic salmon (*Salmo salar*) [10] and Arctic char (*Salvelinus alpinus*) [11]. Gut microbiota, which facilitate host homeostasis [12, 13], have been analysed in many fish species [14, 15] but rarely in the hilsa fish [5]. The gut microbiota of fish is recognized for its pivotal roles in several fundamental aspects, including digestion, nutrition, immunity, reproductive functions, and overall health maintenance in fish [16]. In a previous study, *Aeromonas*, *Pseudomonas*, *Vibrio*, *Streptococcus* and other coliforms have been found in the intestine of fish [5]. The presence of different genera in the gut microbiome of Atlantic salmon from freshwater and marine water indicates the adaptability of the fish's microbiota to its environment. For instance, *Ruminococcus*, *Mycoplasma* and *Pseudomonas* were identified as the predominant genera in the gut of Atlantic salmon from freshwater [17] and *Leuconostoc*, *Vibrio*, *Aliivibrio*, *Photobacterium* and *Weissella* from marine water [18]. In a recent study on hilsa microbiome, *Leucobacter* in gut and *Serratia* in skin mucus were identified as the core bacterial genera, while *Acinetobacter*, *Pseudomonas* and *Psychrobacter* exhibited differential compositions in gut, skin mucus and water [2]. In addition to resident microbes, the fish gut is also regarded as the primary habitat for the colonization of pathogenic bacteria [19]. Studying the diversity of the gut microbiota in fish, such as hilsa, is crucial for understanding its role in the host's health, nutrition, and overall well-being. The gut microbiome is subject to influence from a multitude of factors, making it challenging to determine the specific individual effects of each factor [20]. The colonization of the

fish gut begins at an early stage, during the larval phase, and progresses continuously toward the establishment of a diverse and intricate assembly of gut-associated microbes [21]. Furthermore, inadequate hygiene practices during handling, substandard preservation techniques, and improper transportation methods might render this fish susceptible to becoming a potential carrier for transmitting pathogenic bacteria. In a previous study conducted by Foysal et al., various pathogenic bacteria were identified and characterized in hilsa fish sold in markets across Bangladesh [5]. The majority of research efforts in fish microbiota have concentrated on studying the microbiota associated with the gut and its crucial role in regulating the physiology and health of the host. This emphasis has been particularly notable in aquaculture fish species [22].

Anadromous fishes are known for their migrations between freshwater and marine environments, and the hilsa is no exception. Therefore, understanding the feeding habits of the hilsa throughout various life phases is crucial for the sustainable management and protection of the hilsa fishery [23]. Furthermore, habitats with variable conditions may exert selective pressures on the gut bacteriome, leading to the development of a more resilient microbial community [24]. Fish in habitats with greater environmental variability might host a gut microbiome more capable of maintaining stability despite fluctuating conditions. However, there is a lack of information on the gut microbiomes of hilsa fish in different habitats. Therefore, we hypothesize that hilsa fishes inhabiting different habitats (e.g., freshwater, brackish water, marine) are likely to host distinct microbial communities in response to these environmental variations. In this study, we used the high-throughput 16S rRNA gene-based amplicon sequencing technique to delve into the composition and diversity of the gut bacteriomes in hilsa fish residing in three primary habitats across Bangladesh. Our investigation, based on culture-independent 16S rRNA gene amplicon sequencing, unveiled unique gut bacteriome diversity which indicated that both the signature and diversity of the bacteriome might differ based on the habitat of the hilsa fish.

## 2. Materials and methods

### 2.1. Ethical statement

The Animal Research Ethics Committee (AREC) at Bangabandhu Sheikh Mujibur Rahman Agricultural University, Bangladesh, thoroughly reviewed and approved the experimental procedures conducted in this study (Reference number: FVMAS/AREC/2023/6679). The fish sampling protocol strictly adhered to the ARRIVE 2.0 guidelines [25]. The study adhered to the guidelines outlined in the "Guide for the Care and Use of Laboratory Animals" provided by the National Institutes of Health. The research location was not privately owned or protected, and the study did not involve the use of endangered or protected species.

### 2.2. Sample collection, processing and genomic DNA extraction

Fifteen hilsa fishes (N = 15) were procured from three primary habitats in Bangladesh during the period between September and October 2021 (S1 Table in S1 File). Local fishermen captured the fishes using gillnets, after which the specimens were promptly placed on ice (at 4˚C) and transported to the laboratory at Bangabandhu Sheikh Mujibur Rahman Agricultural University (BSMRAU) within 12 hours for further handling and processing (S1a –S1c Fig in S1 File). The digestive tract from the stomach to the hindgut was removed intact. The gut contents (100 mg) of these fishes were squeezed out from the stomach to the hindgut, and stored in sterile 15 ml centrifuge tubes at -80˚C until DNA extraction (S1d–S1f Fig in S1 File). DNA extraction from each specimen was performed using the DNeasy Blood and Tissue Kit (Qiagen, Crawley, UK) according to the manufacturer's instructions and previously established

protocols [13, 26]. The purity and concentration of the extracted DNA were measured using a NanoDrop 2000c spectrophotometer (Thermo-Fisher Scientific, Waltham, MA).

### 2.3. Library preparation, sequencing and bioinformatics analysis

The genomic DNA was amplified by focusing on the V3-V4 regions of the 16S rRNA gene. The amplification process used a final volume of 30 μL, comprising 3 μL of template DNA, 15 μL of master mix (BioLabs, USA), and 1.5 μL each of the V3-V4 forward (341f: 5′–CCTACGGGNGGCWGCAG–3′) and reverse (785r: 5′–GACTACHVGGGTATCTAATCC–3′) primers [27], and 9 μL ddH2O. The amplification process involved 25 cycles of amplicon PCR, commencing with an initial denaturation step at 95˚C for 3 minutes. Subsequently, each cycle included denaturation at 95˚C for 30 seconds, primer annealing at 55˚C for 30 seconds, and elongation at 72˚C for 30 seconds. Finally, a final extension step was conducted at 72˚C for 5 minutes to complete the process in a thermal cycler (Analytik Jena, Germany) [28]. The PCR amplicons were observed or visualized using a 1.5% agarose gel. Following this visualization, the amplified PCR products underwent purification using Agencourt Ampure XP beads (Beckman Coulter, Brea, USA). For the paired-end library preparation, the Nextera XT index kit (Illumina, San Diego, USA) was utilized. The library preparation process followed the standard protocol provided by Illumina (Part# 15044223 Rev. B). The prepared library pools underwent paired-end (2 × 150 bp) sequencing using the Illumina NextSeq 550 platform (Illumina, USA) at the Illumina Genome Sequencing laboratory of the Institute of Biotchnology and Genetic Engineering (IBGE) of BSMRAU. The quality control and preprocessing of the sequencing data involved the use of FastQC v0.11.9 [29] to assess the quality and Trimmomatic v0.39 [30] with specific parameters (leading: 20, slidingwindow: 4:20:20, trailing: 20, minlen = 36) [13] to remove Illumina adapters, known Illumina artifacts, and phiX reads. The demultiplexed sequences were processed using QIIME 2 (2023.2.0) along with its associated plugins [31]. The SILVA database v.138 [32] was exploited to assign these processed sequences into operational taxonomic units (OTUs) based on a similarity threshold of ≥ 98%. For most bioinformatic analyses, default parameters were employed, unless specific modifications were made.

### 2.4. Statistical analysis

The R programming language (v4.1.1), was employed for downstream analysis, encompassing tasks such as alpha and beta diversity assessments, microbial composition analysis, and statistical comparisons. To estimate the diversity within samples (α-diversity), several diversity indices including observed OTUs, Chao1, Shannon, Simpson, InvSimpson, and Fisher diversity indices were calculated. These calculations were performed using the microbiomeSeq package (http://www.github.com/umerijaz/microbiomeSeq). Taxa abundances were normalized by Total Sum Scaling (TSS) that uses the total read count for each sample as the size factor [33]. Additionally, the visualization of these diversity indices was carried out using the phyloseq R package (v1.34.0) [34]. The differences in bacterial diversity across the various host habitats were assessed using the non-parametric Kruskal-Wallis test. This statistical test is particularly suitable for comparing multiple groups or habitats when analysing non-normally distributed data [35].

## 3. Results

### 3.1. Gut bacteriome diversity and composition in hilsa fishes

To uncover the diversity and signature of the bacteriome in the gut of hilsa fishes obtained from three distinct habitats in Bangladesh (FW: freshwater, BW: brackish water, MW: marine

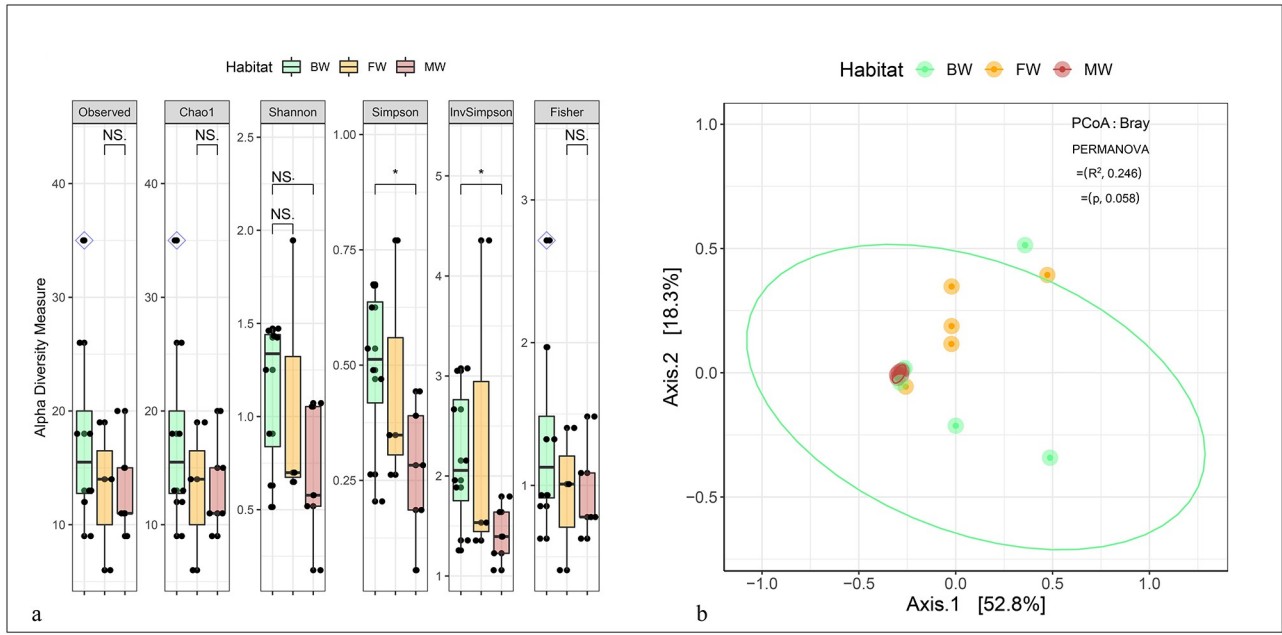

**Fig 1. Bacteriome diversity in hilsa fish.** (a) Within subject (Alpha) diversity measure. Observed, Chao1, Shannon, Simpson, InvSimpson and Fisher indices estimated alpha diversity in hilsa fish samples according to host habitat i.e., freshwater (FW), brackish water (BW) and marine water (MW). The within sample diversity are plotted on boxplots and comparisons are made with pairwise Wilcoxon rank sum tests. Significance level (p-value) 0.01 and 0.05 are represented by the symbols "**", and "*", respectively. (b) Between subject (Beta) diversity measure according to host habitat i.e., FW, BW and MW. Bacterial beta diversity was calculated using Bray-Curtis dissimilarity distance method, and visualized on principal coordinate analysis (PCoA) plots. The samples are coloured according to host habitat (e.g., FW: cheese orange, BW: dragon green and MW: cherry red) and joined with the respective ellipses. Pairwise comparisons on a distance matrix using PERMANOVA test under reduced model shows significant bacterial community differences across the habitats ($p = 0.05$, $R^2 = 0.246$). NS refers to non-significant.

water), we conducted an analysis of 15 gut samples (intestinal contents) using 16S rRNA gene-based amplicon sequencing. Detailed information regarding the study's sampling, demographics, amplicon sequencing data, assigned operational taxonomic units (OTUs) per sample, and the corresponding SRA (Sequence Read Archives) accession numbers for the study subjects are summarized in S1 Table in S1 File. The mean length, girth, and weight of the hilsa fishes were recorded as 39.77 ± 6.13 cm, 25.07 ± 4.91 cm, and 827.40 ± 263.99 grams, respectively (S2 Table in S1 File). The 16S rRNA gene amplicon sequencing conducted on the 15 hilsa fish samples resulted in the generation of 3,234,208 raw reads, averaging approximately 215,614 reads per sample. Out of these, 380,330 quality reads (representing 11.76% of the total) were successfully mapped to identify 258 operational taxonomic units (OTUs) of bacteria (S1 Table in S1 File).

The investigation aimed to assess whether the bacteriome diversity of hilsa fishes differed across various habitats (FW, BW, and MW) by examining both within-sample (alpha) and between-sample (beta) diversities of the detected bacterial communities (Fig 1). Alpha diversity, measured by Observed species, Chao1, Shannon, Simpson, InvSimpson, and Fisher indices, demonstrated notable differences in bacterial community richness. The gut of hilsa fishes collected from FW exhibited significantly higher diversity, followed by BW and MW ($p = 0.05$; Wilcoxon test) (Fig 1a). The analysis utilizing Bray—Curtis dissimilarity distance, leading to a principal coordinate analysis (PCoA) plot, displayed significant variations in bacteriome composition among hilsa fishes' gut samples from distinct habitats ($p = 0.05$, R2 = 0.246, PERMANOVA test) (Fig 1b). The observed OTUs consisted of six phyla, nine classes, 19 orders, 26

families, and 40 genera of bacteria (S2 Table in S1 File). Venn diagrams (S2 Fig in S1 File) illustrated the distribution of unique and shared bacterial taxa between DF and NDF samples. Among the six bacterial phyla detected, 66.67% (4/6) were found to be shared across FW, BW, and MW samples (S2a Fig in S1 File). At the class level, nine bacterial classes were identified, with 77.78% (7/9) being shared among the hilsa fish habitats (S2b Fig in S1 File, S1 Data). Regarding habitats (S2c Fig in S1 File), while 61.54% (16/26) of families and 67.50% (27/40) of genera remained shared across FW, BW, and MW samples. Notably, one bacterial genus in FW and one genus in BW showed exclusive association (S2e Fig in S1 File, S1 Data).

## 3.2. Taxonomic composition of bacteriomes in the gut of hilsa fishes

The microbiome analysis conducted on hilsa fishes from three distinct habitats (FW, BW, and MW) showcased notable differences among their gut bacterial communities. At the phylum level, Firmicutes, Proteobacteria, and Planctomycetes predominated, accounting for over 97.5% of the total abundances (S1 Data). Comparing the relative abundances of these phyla across the hilsa fish habitats revealed distinct patterns. Proteobacteria was notably abundant in FW (90.78%) and BW (93.17%), while Firmicutes dominated in MW, representing an average relative abundance of 92.03%. Moreover, Firmicutes ranked as the second most abundant phylum in FW (6.71%) and BW (4.82%), whereas Proteobacteria remained as the second most predominant phylum in MW (7.76%). The relative abundances of other phyla also significantly differed across the hilsa fish gut habitats (Fig 2, S1 Data). The major bacterial classes detected in the hilsa fish gut were *Bacilli* (67.84%), *Gammaproteobacteria* (22.25%), *Synechococcophycideae* (5.50%), and *Clostridia* (2.35%). These classes exhibited substantial variations in relative abundances across the three habitats. *Bacilli* prevailed as the top abundant class in FW (82.50%) and MW (90.43%). Conversely, *Gammaproteobacteria* dominated in BW (79.68%) followed by *Bacilli* (17.50%). Additionally, *Gammaproteobacteria* appeared as the second most abundant class in FW (15.09%) and MW (9.28%). The relative abundances of other bacterial classes remained comparably lower (< 2.0%) and varied across the three habitats (S1 Data).

By comparing the bacterial taxa at order level, we found that *Enterobacteriales* (92.08%), *Clostridiales* (4.11%) and *Lactobacillales* (1.97%) were the top bacterial orders in the gut of hilsa fishes of FW whereas *Enterobacteriales* (94.08%) and *Lactobacillales* (3.59%) in BW, and *Lactobacillales* (92.25%) and *Enterobacteriales* (7.56%) in MW were the predominant orders in the gut of hilsa (Fig 3, S1 Data). We also found significant differences ($p$ = 0.03, Kruskal-Wallis test) in the composition and relative abundance of the gut bacteria at the family level (S3 Fig in S1 File, S1 Data). In the hilsa fish of FW, *Enterobacteriaceae* was the most abundant (90.12%) bacterial family followed by *Clostridiaceae* (2.17%), *Moraxellaceae* (2.10%), *Peptostreptococcaceae* (1.95%) and *Enterococcaceae* (1.50%). Likewise, *Enterobacteriaceae* (85.10%), *Peptostreptococcaceae* (3.61%), *Enterococcaceae* (3.13%), *Clostridiaceae* (2.15%), *Planctomycetaceae* (2.14%), *Staphylococcaceae* (1.27%) and *Gemmataceae* (1.12%) in BW, and *Enterococcaceae* (89.69%) and *Enterobacteriaceae* (8.62%) in MW were the top abundant bacterial families (S3 Fig in S1 File). Although, the relative abundance of the rest of the bacterial orders and families were substantially lower (< 1.0%), they showed marked discriminations both in composition and abundances according to the habitats of the hilsa fishes (S1 Data).

The study showcased significant differences in both the composition and relative abundances of bacterial taxa at the genus level across different habitats (FW, BW, and MW) of the hilsa fishes ($p$ = 0.007; Kruskal Wallis test). A total of 40 bacterial genera were detected in the gut of hilsa fishes, with 36, 37, and 32 genera identified in FW, BW, and MW, respectively (S2 Table in S1 File). Among these, 67.5% were shared among hilsa fishes from all three habitats

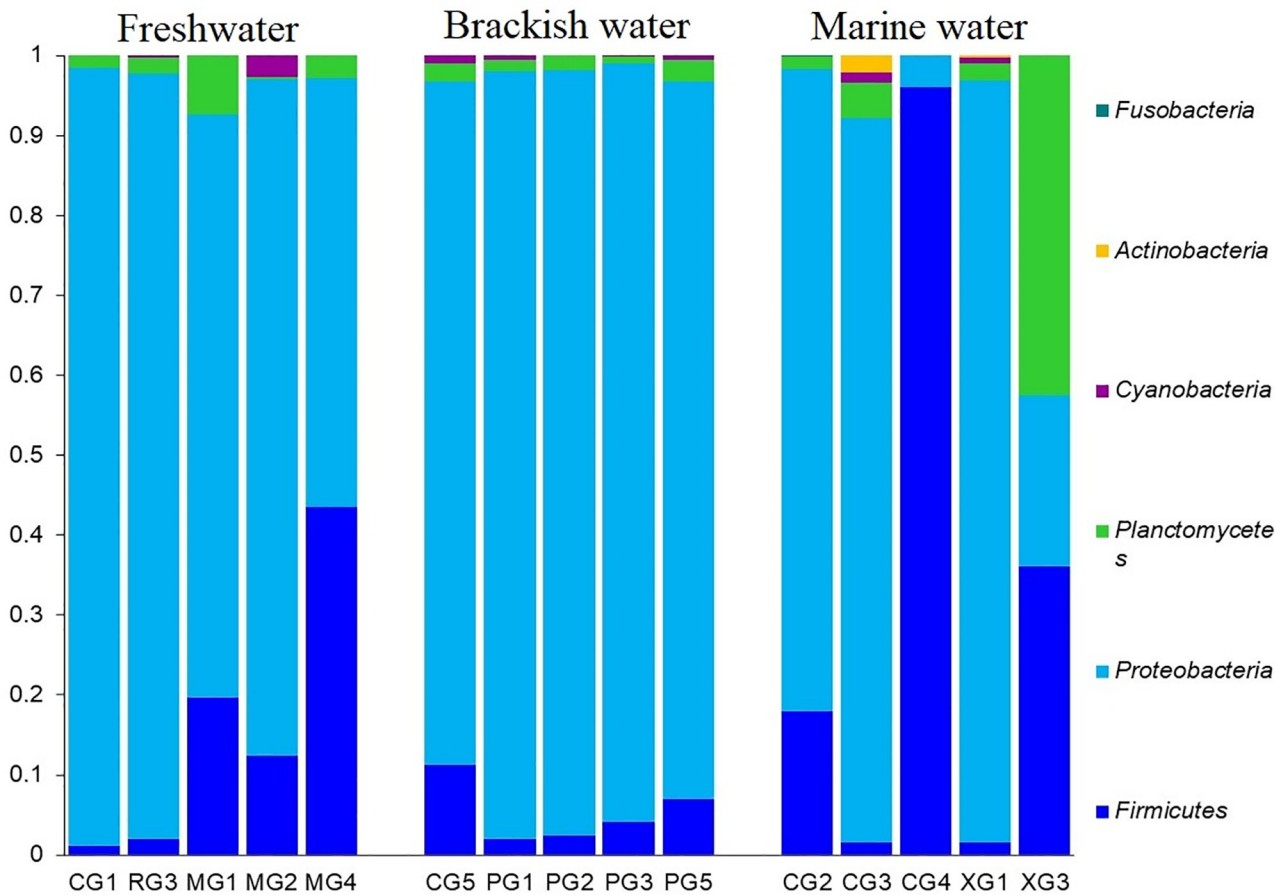

**Fig 2. Taxonomic profile of bacteriomes in hilsa fish at the phylum level.** The bar plots representing the relative abundance of bacterial phyla in freshwater (FW), brackish water (BW) and marine water (MW) samples. Each stacked bar plot represents the abundance of bacterial phyla in each sample of the corresponding category. Notable differences in bacterial phyla are those where the taxon is abundant in one habitat, and effectively not detected in rest of the two habitats. The distribution and relative abundance of the bacterial phyla in the study metagenomes are also available in S1 Data.

(S2e Fig in S1 File). By exploring the phylogenetic relationships of the identified bacterial genera, it was observed that the majority (n = 18) belonged to *Gammaproteobacteria*, followed by *Bacilli* (n = 15), *Actinobacteria* (n = 2), *Planctomycetia* (n = 2), and other miscellaneous groups (n = 3) (S4 Fig in S1 File). The gut bacteriome of BW hilsa fishes exhibited dominance by genera such as *Morganella* (21.88%), *Plesiomonas* (16.86%), *Enterobacter* (15.81%), *Proteus* (10.36%), *Vagococcus* (8.79%), *Serratia* (5.36%), *Cronobacter* (4.96%), *Klebsiella* (4.56%), *Clostridium* (3.67%), *Shigella* (2.36%), and *Acinetobacter* (2.33%). Conversely, *Vagococcus* (83.49%), *Enterobacter* (5.31%), *Morganella* (5.09%), and *Shigella* (1.11%) were predominant in the gut of MW hilsa fishes (Fig 4). Several identified bacterial genera showed significant correlations ($p < 0.05$, Kruskal Wallis test) within specific hilsa fish habitats. For instance, *Acinetobacter*, *Klebsiella*, *Pseudomonas*, and *Planctomyces* in FW, *Enterobacter*, *Morganella*, *Klebsiella*, and *Shigella* in BW, and *Proteus* in MW hilsa fishes displayed significant correlations. Notably, *Lactococcus*, *Macrococcus*, and *Vagococcus* exhibited stronger correlations ($p < 0.05$, Kruskal Wallis test) across all samples from the three habitats (FW, BW, and MW) (Fig 5). Interestingly, six out of the 40 identified bacterial genera (*Sinobaca*, *Synechococcus*,

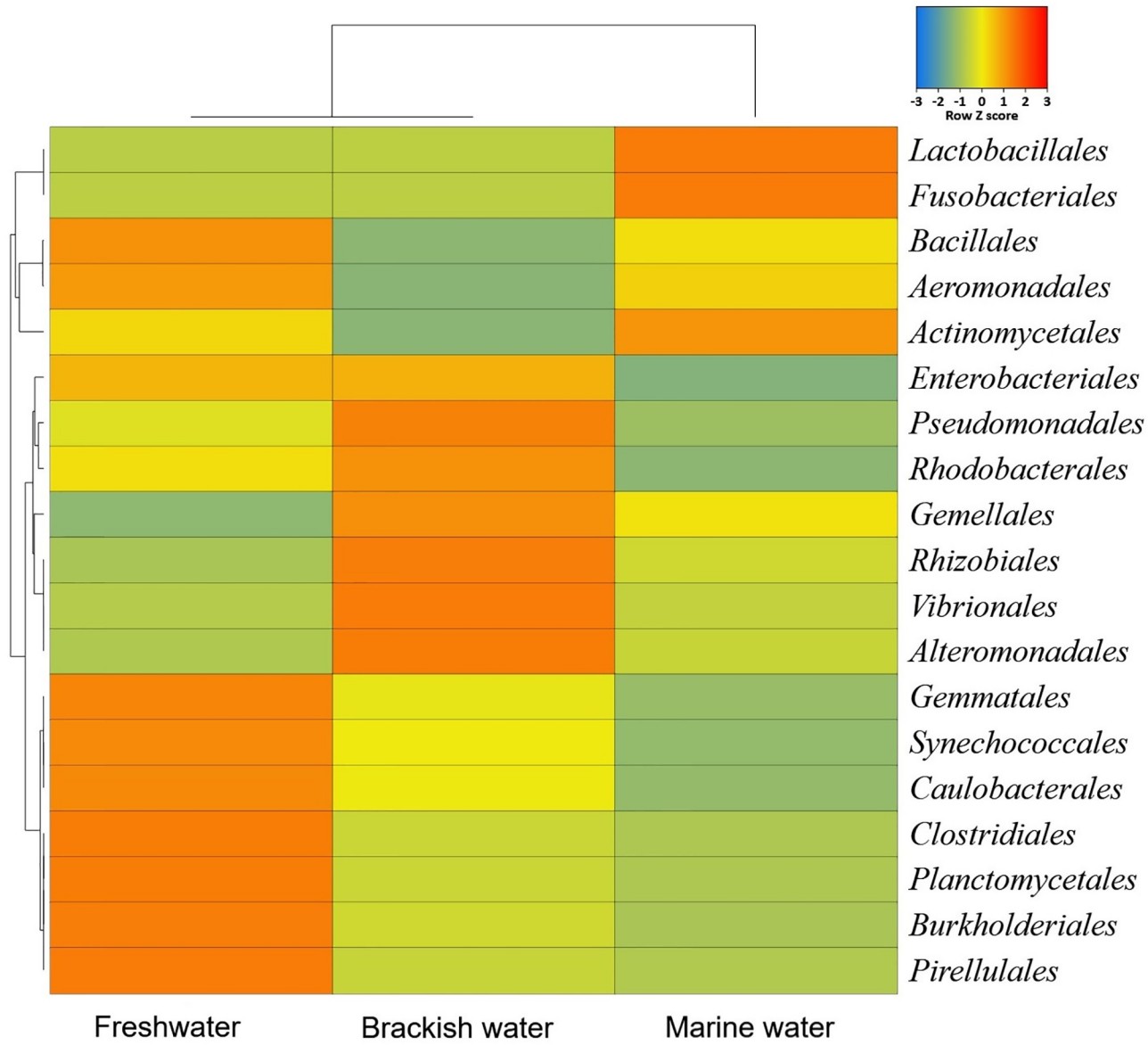

**Fig 3. Taxonomic profile of bacteriomes in hilsa fish at the order level.** Heatmap showing the average relative abundances and hierarchical clustering of the bacterial orders in the study in freshwater (FW), brackish water (BW) and marine water (MW) samples. The colour bar (row Z score) at the top represents the relative abundance of the bacterial orders in the corresponding samples. The colour codes indicate the presence and completeness of each bacterial taxa, expressed as a value between = 3 (lowest abundance) and 3 (highest abundance). The red colour indicates the more abundant patterns, while blue cells account for less abundant bacterial orders in that particular sample.

*Gemmata*, *Serinicoccus*, *Saccharopolyspora*, and *Paulinella*) had not been previously identified in any aquatic and marine fish species. Although the relative abundances of the remaining genera were low (<1.0%), they showed differences across the three hilsa fish habitats (FW, BW, and MW) (Fig 4, S1 Data).

## 4. Discussion

Studying the gut microbiome of hilsa fish (*T. ilisha*) is crucial for understanding its ecological dynamics, potential impacts on fish health, and its role in maintaining ecosystem balance

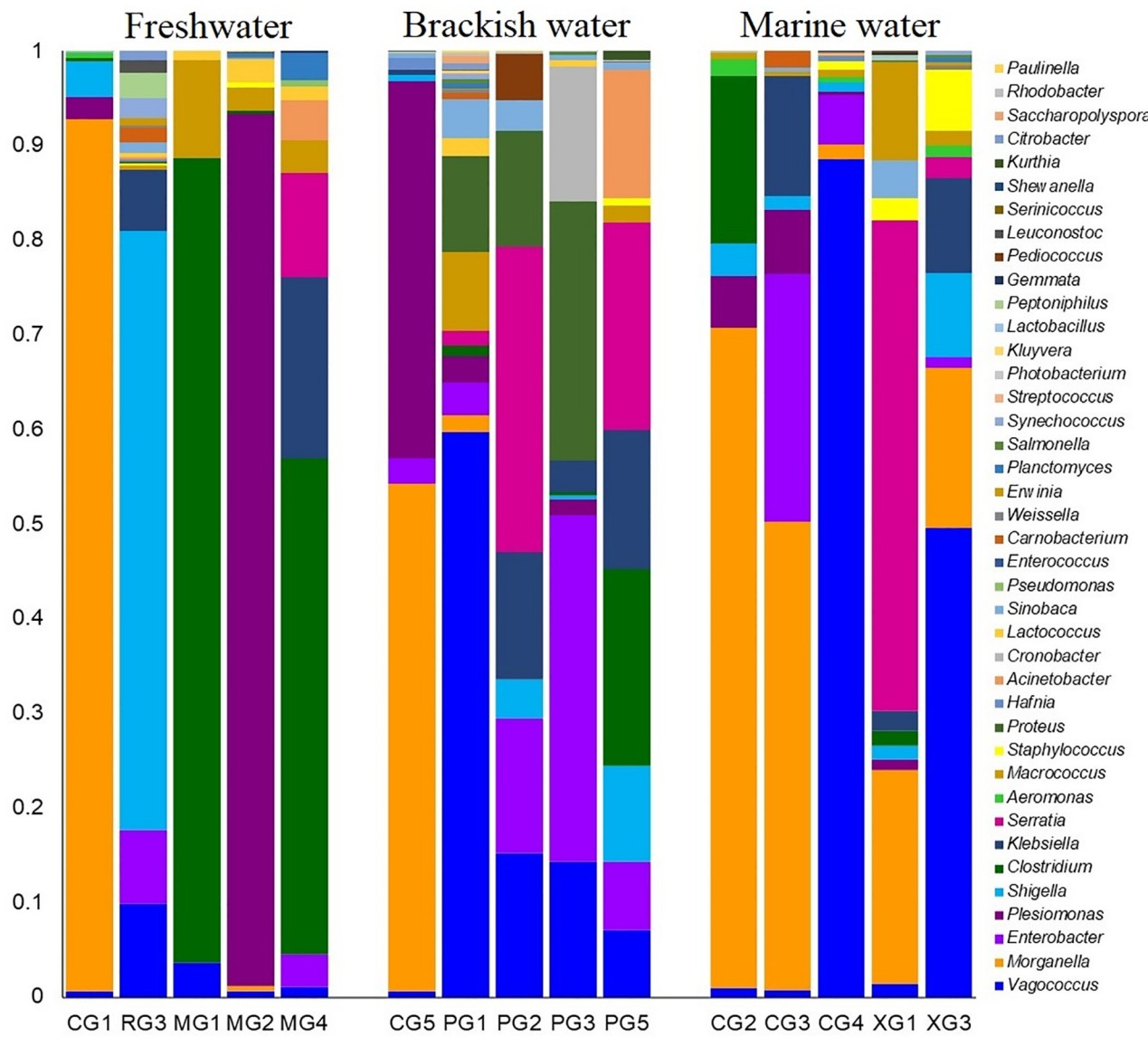

**Fig 4. The genus-level taxonomic profile of bacteriomes.** The bar plots representing the relative abundance of 40 bacterial genera in freshwater (FW), brackish water (BW) and marine water (MW) samples. Each stacked bar plot represents the abundance of bacterial genera in each sample of the corresponding category. Notable differences in bacterial genera are those where the taxon is abundant in one habitat, and effectively not detected in rest of the two habitats. The distribution and relative abundance of the bacterial genera in the study metagenomes are also available in S1 Data.

within aquatic environments. The goal of this study was to analyze the microbial communities in the diverse habitats of the national fish of Bangladesh, *T. ilisha*, also known as hilsa, using next-generation sequencing (NGS) techniques. We employed, a targeted approach using amplicon sequencing of 16S rRNA gene (V3-V4 regions) sequencing-based metagenomics to identify diverse bacterial taxa and their distribution within the gut of hilsa fishes collected from three distinct habitats (freshwater; FW, brackish water; BW, and marine water; MW) in Bangladesh. This study represents one of the groundbreaking efforts to explore the gut micro-biota of wild hilsa fish, offering comprehensive insights into the composition and relative abundances of the gut bacteriomes across major habitats in Bangladesh.

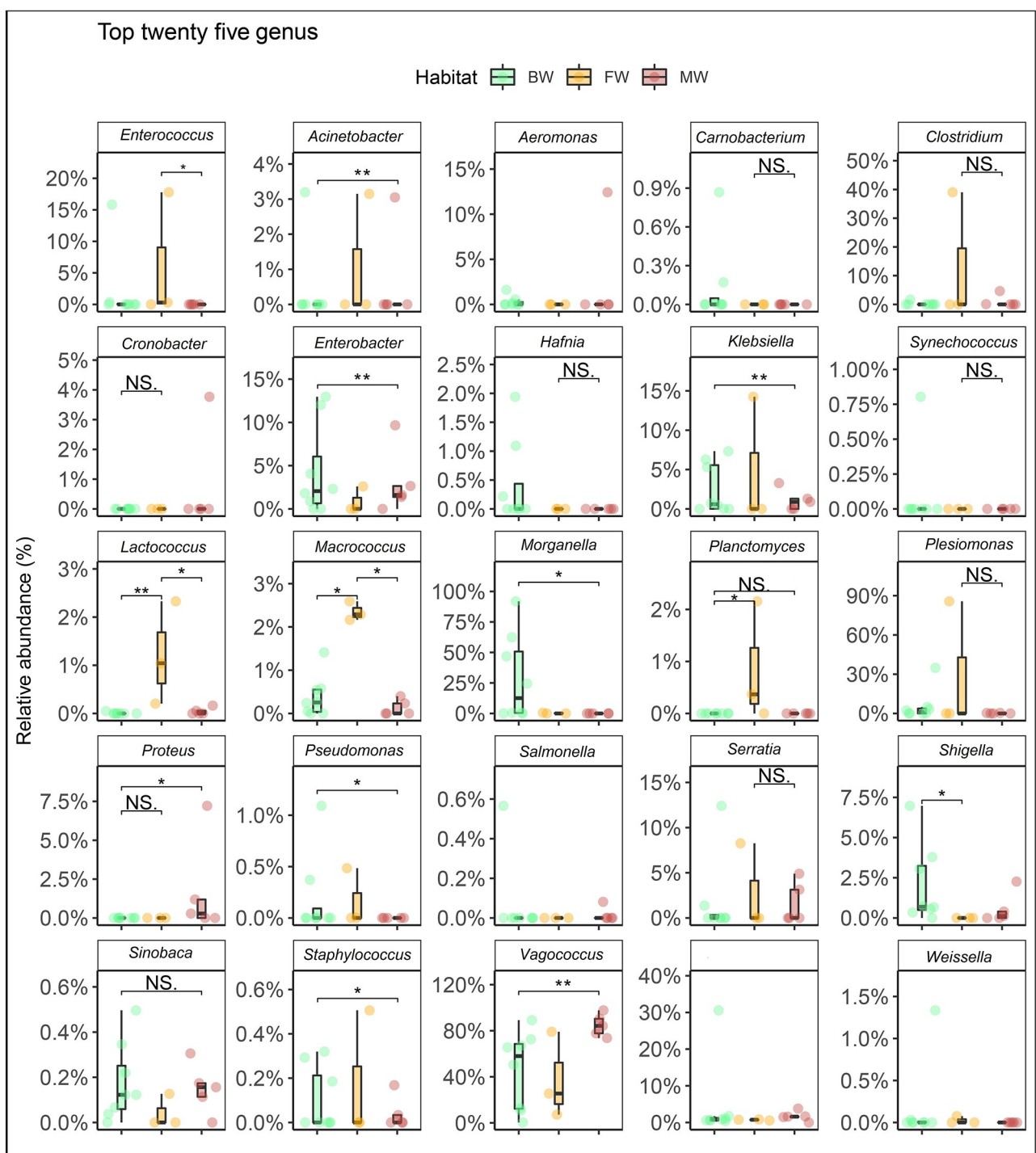

**Fig 5. Mean relative abundance and within-group divergence of top abundant 25 bacterial genera.** Relative abundances of top 25 abundant genera and within-group (i.e., freshwater; FW, brackish water; BW and marine water; MW) divergence by use of Bray—Curtis dissimilarity. In the boxplots, the middle line represents the median, the lower hinge corresponds to the first quartile (25th percentile), the upper hinge corresponds to the third quartile (75th percentile), the whiskers extend to the largest and smallest values. The samples are coloured according to host habitats (e.g., FW: cheese orange, BW: dragon green and MW: cherry red). Pairwise comparisons were done by use of the Kruskal—Wallis test. All *p* values are adjusted. We considered *p* values less than 0.05 and adjusted *p* values less than 0.1 to be significant. NS refers to non-significant.

We accurately classified taxonomy of bacteriome by employing high-throughput 16S rRNA gene amplicon sequencing and advanced bioinformatics, utilizing thousands of assigned reads. Notably, the three most prevalent and abundant phyla observed in this research (Firmicutes, Proteobacteria, and Planctomycetes) have been previously documented in carriage water of ornamental fish and commercially important fishes across various continents [36, 37]. In a recent study, Biswas et al. reported that > 60% of the identified bacteria were Proteobacteria with modest abundance (> 5%) of Firmicutes, Bacteroidetes and Actinobacteria in the gut of hilsa fishes [2]. Furthermore, a significant finding of this study is that the majority of bacterial phyla identified, regardless of the hilsa fish habitats, have been documented to constitute a substantial portion of the gut microbiota in numerous fish species. This observation further supports the relevance and consistency of these bacterial phyla across diverse fish populations, indicating their probable important roles in fish gut ecosystems [16, 17, 38].

A key finding in this study was the discovery of multiple potentially beneficial and/or probiotic bacterial genera within the gut of hilsa fish across the three distinct habitats. Among the 40 genera detected, *Vagococcus*, *Morganella*, *Serratia*, *Enterobacter*, *Aeromonas*, and *Klebsiella* emerged as the dominant genera observed in the hilsa fish gut. Notably, the genus *Vagococcus* encompasses 13 species, with the majority of these species being recently recognized and utilized within aquaculture as promising candidates for probiotics in marine fish species. This finding suggests the potential significance of these bacterial genera in contributing to the health and ecological balance within the gut environment of hilsa fishes across varied habitats [39]. Certainly, while *Vagococcus* spp. have shown promise as a probiotic agent in aquaculture, concerns about its safety highlight that absolute risk elimination cannot be guaranteed. *Morganella*, a Gram-negative, rod-shaped bacterium, is commonly found in freshwater environments, soil, and within the normal flora of animal intestinal tracts, including fish [40], as well as in fish products [41]. Meanwhile, *Enterobacter* spp. has been demonstrated as a potential probiotic for aquaculture, exhibiting extensive resistance to bacterial infections, lacking pathogenicity toward the host, and showcasing robust environmental tolerance [42]. In a recent study by Tang et al., *E. asburiae* C28, isolated from the intestine of *Carassius auratus*, displayed promising attributes [43]. It notably reduced the load of potential pathogens, increased the population of potential probiotics within the host gut, and decreased the mortality rate of *C. auratus* when challenged by *A. hydrophila* [43]. These findings underscore the potential of certain bacterial species, such as *Enterobacter* spp., in enhancing fish health and disease resistance, contributing to the development of probiotic strategies in aquaculture. However, continuous evaluation and cautious consideration regarding the safety and efficacy of probiotics in aquaculture practices remain essential. Indeed, beyond the previously mentioned genera, a variety of probiotic bacterial genera, such as *Plesiomonas*, *Pseudomonas*, *Acinetobacter*, *Streptococcus*, and *Staphylococcus*, were detected within the gut content of hilsa fishes across three significant habitats in Bangladesh.

*Plesiomonas*, a genus known to inhabit both freshwater and marine environments, has been previously isolated from various aquatic and marine species. This genus represents a versatile group of bacteria that demonstrate adaptability to diverse aquatic environments, further highlighting their potential importance within the gut ecosystem of hilsa fishes in different habitats [44, 45]. Certainly, *Aeromonas* species exhibit a wide distribution across various aquatic and environmental habitats, including freshwater, sediment, estuaries, seaweed, sea grass, used water, drinking water, and even within food sources [46]. Notably, both *Aeromonas* sp. and *Pseudomonas* sp. have been identified as predominant bacteria frequently isolated from carp culture systems [47]. In the aquaculture industry, numerous microbial species, particularly Gram-negative bacteria like *Aeromonas*, *Enterobacter*, *Pseudoalteromonas*, *Pseudomonas*, *Acinetobacter*, *Rhodopseudomonas*, and *Vibrio*, alongside Gram-positive bacteria such as

*Bacillus*, *Enterococcus*, *Lactobacillus*, *Lactococcus*, *Microbacterium*, *Micrococcus*, *Streptococcus*, and *Streptomyces*, have been reported and introduced as probiotics [48, 49]. Indeed, the isolation and efficacy of probiotic bacteria derived from hilsa fish have not been documented or reported. Additionally, a noteworthy discovery from this study was the identification of certain bacterial genera (such as *Sinobaca*, *Synechococcus*, *Gemmata*, *Serinicoccus*, *Saccharopolyspora*, and *Paulinella*) that have not been previously reported in any aquatic or marine fish species. However, it's noteworthy that some of these genera, including *Synechococcus*, *Gemmata*, *Saccharopolyspora*, and *Paulinella*, have been identified as part of photosynthetic ocean microbiomes thriving in diverse salinity conditions and varying seasons. The presence of these unique bacterial genera in the gut microbiota of hilsa fishes indicates the potential influence of various environmental factors on the composition and diversity of their gut bacteriomes. Their association suggests the intricate interplay between the fish's habitat, environment, and the microbial communities inhabiting their gastrointestinal tracts, potentially contributing to the overall ecology of the hilsa fish and its habitat [50, 51]. Furthermore, the identification of bacterial genera like *Lactococcus*, *Morganella*, *Enterococcus*, *Aeromonas*, *Shewanella*, *Pediococcus*, *Leuconostoc*, *Saccharopolyspora*, *and Lactobacillus* in this study holds significance as these genera are frequently recognized and utilized as common probiotic bacteria in aquaculture industries. Their presence within the gut microbiota of hilsa fishes across different habitats suggests potential functional roles in maintaining gut health, aiding digestion, modulating immunity, and possibly contributing to the overall well-being of these fish species. The utilization of these bacteria as probiotics in aquaculture further highlights their importance and potential benefits in enhancing fish health and promoting sustainable aquaculture practices [36, 50, 52, 53]. Furthermore, the involvement of probiotics in enhancing various aspects of fish health and well-being has been extensively documented across different fish species.

In the current study, the probiotic microbes identified within the gut microbiota of hilsa fishes may indeed play crucial roles in enhancing several aspects related to their health and quality. These probiotic microbes potentially contribute to enhancing the nutritional value, flavor, and texture of hilsa fish. Additionally, they might actively produce antioxidant and antimicrobial compounds, thereby aiding in preserving the quality and freshness of the fish and potentially extending its shelf life. Moreover, these probiotics can also stimulate and modulate immune functions in hilsa fishes, bolstering their disease resistance mechanisms and overall immune response. The multifaceted roles of these probiotic microbes within the gut of hilsa fishes underscore their potential significance in improving the fish's quality, health, and potential benefits for consumers.

Our study is constrained by a relatively small sample size (n = 5) per group of hilsa fishes in three habitats, making it challenging to generalize findings to broader hilsa populations or different geographical regions. Addressing this limitation requires future research with a larger sample size, employing robust experimental design and statistical analysis for a more comprehensive understanding of microbiome diversity and composition in hilsa fishes across varied geographic habitats. Exploring differences in gut bacteriome composition across habitats may reveal correlations with functional variations in microbial metabolic pathways, gene expression related to nutrient metabolism, detoxification, or host-microbe interactions.

## 5. Conclusion

The gut microbiome of fish plays a crucial role in various physiological functions including digestion, immune system development, and nutrient absorption. High-throughput 16S rRNA gene sequencing, analysed with advanced bioinformatics tools, has uncovered insights into the gut bacteriome of hilsa fish across freshwater, brackish water, and marine habitats, facilitating a deeper understanding of this economically significant anadromous species' microbiome.

The hilsa fish, being an important transboundary species dwelling in the Bay of Bengal and exhibiting migratory behaviours to upstream rivers in Bangladesh and neighboring South Asian countries for various life cycle stages such as feeding, breeding, and nurturing offspring, displays a clear influence of its habitats on the composition of its microbiota. The findings from this study yield valuable insights into the composition, structure and distribution of gut microbiomes of hilsa fish. Interestingly, we discovered *Vagococcus*, *Morganella*, *Enterobacter*, *Aeromonas*, *Pseudomonas*, *Plesiomonas*, *Acinetobacter*, *Enterococcus*, *Lactobacillus*, *Lactococcus*, *Streptococcus*, among others as potential beneficial probiotic bacterial genera. Importantly, some bacterial genera such as *Sinobaca*, *Synechococcus*, *Gemmata*, *Serinicoccus*, *Saccharopolyspora*, and *Paulinella* identified in the gut of hilsa identified in this study have not been reported in any aquatic or marine fish species. These findings hold potential implications for prospective applications in aquaculture practices. Collectively, the data obtained from the analysis of the hilsa fish's bacteriome and taxonomic observations reported in this study lay a strong foundation for further, more comprehensive investigations. Future studies with a larger sample size can delve deeper into understanding the co-evolutionary aspects of hilsa fish microbiomes, particularly focusing on elucidating the roles played by the gut and flesh microbiota in host metabolism, immunity, and overall health. Such in-depth investigations could offer crucial insights into optimizing aquaculture practices and fostering the sustainable management of hilsa fish populations.

## Supporting information

**S1 Data. Taxonomic information on the gut microbiomes of hilsa fish in three habitats (e.g., freshwater; FW, brackish water; BW, and marine water; MW).**
(XLSX)

**S2 Data.**
(XLSX)

**S1 File.**
(DOCX)

## Acknowledgments

The authors would also like to thank Mr. Nur Uddin Mahmud, former MS student of the IBGE, BSMRAU for his support in technical assistance in genomic DNA extraction and polymerase chain reaction (PCR).

## Author Contributions

**Conceptualization:** A. Q. M. Robiul Kawser, M. Nazmul Hoque, Tofazzal Islam.

**Data curation:** A. Q. M. Robiul Kawser, M. Nazmul Hoque, M. Shaminur Rahman.

**Formal analysis:** M. Nazmul Hoque, M. Shaminur Rahman, Tahsin Islam Sakif.

**Funding acquisition:** Tracey J. Coffey, Tofazzal Islam.

**Investigation:** A. Q. M. Robiul Kawser.

**Methodology:** A. Q. M. Robiul Kawser, M. Nazmul Hoque, M. Shaminur Rahman.

**Project administration:** Tracey J. Coffey, Tofazzal Islam.

**Resources:** M. Nazmul Hoque, M. Shaminur Rahman, Tahsin Islam Sakif, Tofazzal Islam.

**Software:** M. Nazmul Hoque.

**Supervision:** Tracey J. Coffey, Tofazzal Islam.

**Validation:** A. Q. M. Robiul Kawser, M. Nazmul Hoque, M. Shaminur Rahman, Tofazzal Islam.

**Visualization:** M. Nazmul Hoque, M. Shaminur Rahman.

**Writing – original draft:** A. Q. M. Robiul Kawser, M. Nazmul Hoque.

**Writing – review & editing:** Tahsin Islam Sakif, Tracey J. Coffey, Tofazzal Islam.

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
