## [Decision Letter · Decision Letter 0]

12 Mar 2024

PONE-D-24-07704Unveiling the gut bacteriome diversity and signature of the Bangladesh national fish hilsa (Tenualosa ilisha)PLOS ONE

Dear Dr. Islam,

Thank you for submitting your manuscript to PLOS ONE. After careful consideration, we feel that it has merit but does not fully meet PLOS ONE’s publication criteria as it currently stands. Therefore, we invite you to submit a revised version of the manuscript that addresses the points raised during the review process.

We look forward to receiving your revised manuscript.

Kind regards,

Bijay Kumar Behera, Ph.D.

Academic Editor

PLOS ONE

Journal Requirements:

https://doi.org/10.1007/s11033-023-08965-6

In your revision ensure you cite all your sources (including your own works), and quote or rephrase any duplicated text outside the methods section. Further consideration is dependent on these concerns being addressed.

“This research was partially supported with funds from ‘BSMRAU Physical Facility and Research Capacity Strengthening Project’ under the Ministry of Education, People’s Republic of Bangladesh for funding this research, Grant No.: BSMRAU/01/2021. The funders had no role in study design, data collection and analysis, decision to publish, or preparation of the manuscript.”

6. PLOS requires an ORCID iD for the corresponding author in Editorial Manager on papers submitted after December 6th, 2016. Please ensure that you have an ORCID iD and that it is validated in Editorial Manager. To do this, go to ‘Update my Information’ (in the upper left-hand corner of the main menu), and click on the Fetch/Validate link next to the ORCID field. This will take you to the ORCID site and allow you to create a new iD or authenticate a pre-existing iD in Editorial Manager. Please see the following video for instructions on linking an ORCID iD to your Editorial Manager account: https://www.youtube.com/watch?v=_xcclfuvtxQ.

7. Please be informed that funding information should not appear in the Acknowledgments section or other areas of your manuscript. We will only publish funding information present in the Funding Statement section of the online submission form. Please remove any funding-related text from the manuscript.

8. Please include a separate caption for each figure in your manuscript.

10. We are unable to open your Supporting Information file [File Name]. Please kindly revise as necessary and re-upload.

Additional Editor Comments (if provided):

Please find the two reviewers comments, and my decision is a minor revision of the manuscript.

Reviewers' comments:

Reviewer's Responses to Questions

**Comments to the Author**

1. Is the manuscript technically sound, and do the data support the conclusions?

Reviewer #1: Yes

Reviewer #2: Yes

2. Has the statistical analysis been performed appropriately and rigorously? 

Reviewer #1: Yes

Reviewer #2: Yes

3. Have the authors made all data underlying the findings in their manuscript fully available?

Reviewer #1: Yes

Reviewer #2: Yes

4. Is the manuscript presented in an intelligible fashion and written in standard English?

Reviewer #1: Yes

Reviewer #2: Yes

5. Review Comments to the Author

Reviewer #1: The manuscript entitled, Unveiling the gut bacteriome diversity and signature of the Bangladesh national fish hilsa (Tenualosa ilisha) (Manuscript Number: PONE-D-24-07704) by Kawser and co-worker is well written in the manuscript. However, minor modifications are required. Please see the comments.

1. Key Words are to be Keywords

2. The abstract section is well written

3. Line no 147: used instead of utilized

4. The result section is well written

5. Discussion should be more discussed with the latest literature

6. Image quality should to improved

Reviewer #2: The manuscript titled “Unveiling the gut bacteriome diversity and signature of the Bangladesh national fish hilsa (Tenualosa ilisha)” Manuscript Number: PONE-D-24-07704 is well written.

However, refer to the comments below to rectify the manuscript-

1. The title should include “Diversity and distribution” of gut bacteriome.

2. Line No. 45 The Keywords should be Tenualosa ilisha, Gut microbiome, Metagenomics, 16S rRNA, Probiotic properties.

3. In the Introduction, we introduce Tenualosa Ilisha and provide Nutritional information about the Hilsa fish. We also provide the IUCN Status of the particular fish species in view of the conservation effort.

4. Line No. 117 Write the word ‘used’ instead of utilized.

5. The Methodology part of the manuscript is well-written.

6. The discussion part of the manuscript is also well written with distinct functioning of the Bacteriome found.

7. Figure quality is very poor; it should be at least 300 dpi resolution.

6. PLOS authors have the option to publish the peer review history of their article (what does this mean?). If published, this will include your full peer review and any attached files.

Reviewer #1: **Yes: **Ajaya Kumar Rout

Reviewer #2: **Yes: **Sushree Swati Rout

---

## [Author Response · Author response to Decision Letter 0]

26 Mar 2024

March 17, 2024

To,

Bijay Kumar Behera, Ph.D.

Subject: Submission of revised manuscript (R1)

Manuscript ID: PONE-D-24-07704

Manuscript title: Unveiling the gut bacteriome diversity and signature of the Bangladesh national fish hilsa (Tenualosa ilisha)

Dear Editor,

Thank you for the editor's decision letter dated on March 13, 2024. Appended to this letter is our point-by-point responses to the comments raised by both reviewers. We would like to take this opportunity to express our sincere thanks to the expert reviewers/editors who identified several areas in our manuscript that were needed corrections as well as modifications. We also would like to cordially thank you for allowing us the change to resubmit a revised version of the manuscript. 

This research was partially supported with funds from ‘BSMRAU Physical Facility and Research Capacity Strengthening Project’ under the Ministry of Education, People’s Republic of Bangladesh for funding this research, Grant No.: BSMRAU/01/2021. The funders had no role in study design, data collection and analysis, decision to publish, or preparation of the manuscript. There was no additional external funding received for this study.

We have revised and updated the manuscript with some modifications as per reviewers’ suggestion. According to the suggestion of Reviewer 2, we have revised the Title of the manuscript as well (Revised title: Unveiling the gut bacteriome diversity and distribution in the national fish hilsa (Tenualosa ilisha) of Bangladesh). Please find all changes highlighted in RED color fonts in the revised manuscript. We also have provided a clean manuscript for your kind perusal.

Sincerely yours,

Dr. Md Tofazzal Islam, FBAS, FTWAS, FAPS

---

## [Editor Report · Decision Letter 1]

9 Apr 2024

Unveiling the gut bacteriome diversity and distribution in the national fish hilsa (Tenualosa ilisha) of Bangladesh

PONE-D-24-07704R1

Dear Dr. Islam,

We’re pleased to inform you that your manuscript has been judged scientifically suitable for publication and will be formally accepted for publication once it meets all outstanding technical requirements.

Kind regards,

Bijay Kumar Behera, Ph.D.

Academic Editor

PLOS ONE